# HOW (UN)FAIR IS TEXT SUMMARIZATION?

## ABSTRACT

Creating a good summary requires carefully choosing details from the original text to accurately represent it in a limited space. If a summary contains biased information about a group, it risks passing this bias off to readers as fact. These risks increase if we consider not just one biased summary, but rather a biased summarization algorithm. Despite this, no work has measured whether these summarizers demonstrate biased performance. Rather, most work in summarization focuses on improving performance, ignoring questions of bias. In this paper, we demonstrate that automatic summarizers both amplify and introduce bias towards information about under-represented groups. Additionally, we show that summarizers are highly sensitive to document structure, making the summaries they generate unstable under changes that are semantically meaningless to humans, which poses a further fairness risk. Given these results, and the large-scale potential for harm presented by biased summarization, we recommend that bias analysis be performed and reported on summarizers to ensure that new automatic summarization methods do not introduce bias to the summaries they generate.

## 1 INTRODUCTION

In any piece of text, bias against a group may be expressed. This bias may be explicit or implicit and can be displayed either in *what* information is included (e.g., including information that is exclusively negative about one group and exclusively positive about another), *where* in the article it comes from (e.g., only selecting sentences from the start of articles), or *how* it is written (e.g., saying "a man thought to be involved in crime died last night after an officer involved shooting" vs."a police officer shot and killed an unarmed man in his home last night").

The presence of any bias in a longer text may be made worse by summarizing it. A summary can be seen as a presentation of the most salient points of a larger piece of text, where the definition of "salient information" will vary according to various ideologies a person holds. Due to this subjectivity and the space constraints a summary imposes, there is a heightened potential for summaries to contain bias. Readers, however, expect that summaries faithfully represent articles. Therefore, bias in the text of a summary is likely to go unquestioned. If a summary presents information in a way biased against a group, readers are likely to believe that the article exhibited this same bias, as checking the truth of these assumptions requires a high amount of effort. This poses several risks. First, an echo chamber effect, where the bias in generated summaries agrees with biases the reader already has. The opposite is also a risk. An article may present more or less the same amounts of information about multiple groups whereas its summary includes more information about one group, leading readers to believe the most important information in the article is about only one group.

As writing summaries manually carries a large cost, automatic summarization is an appealing solution. However, where one biased summary is a problem, a biased *summarization algorithm* capable of summarizing thousands of articles in the time it takes a human to generate one, is a disaster. In recent years, automatic summarization has increased in availability, both for personal and commercial use. Summarization algorithms have suggested for use on news articles, medical notes, business documents, legal texts, personal documents[1], and conversation transcripts[2]. Despite the sensitivity of these applications, to the best of our knowledge, no work has measured the bias towards groups of summaries generated by common summarization algorithms.

---

[1]https://ai.googleblog.com/2022/03/auto-generated-summaries-in-google-docs.html?m=1
[2]https://learn.microsoft.com/en-us/azure/cognitive-services/language-service/summarization/overview

In this paper, we present the first empirical analysis to measure the bias of automatic summarizers, for seven different techniques (ranging from the ones based on information retrieval methods to the ones using large language models; and for high impact techniques provided by Microsoft and OpenAI). We design a method to quantitatively measure to what extent article structure influences bias in the generated summaries, as well as the inclusion of information about different gender, racial/ethic, and religious groups. We also study the causes of bias by varying the influential factors, including the summarization parameters, and distribution of input documents. We show that summarizers:

1. **Can suppress information about minority group in the original text.**
2. **Prioritize information about certain groups over others, regardless of the amount of information about them in the original text.**
3. **Amplify patterns of bias already shown in human summaries in where information is selected from in the original text.**
4. **Demonstrate sensitivity to the structure of the articles they summarize, and are fragile to changes in those articles that are meaningless to humans.**

These findings indicate that it is not safe to use automatic summarization at scale or in situations where the text they generate could influence large numbers of readers. Doing so risks misinforming and increasing bias held by readers. We conclude that assessments of bias in summarization algorithms should be performed, and this bias should be reduced before models are released.

## 2 WHAT IS BIAS IN A SUMMARIZER?

Let us start by defining what a biased summary is. We consider a biased summary to be be any summary that misrepresents a group in some way, *relative to the original text*[3], whether this is the amount or kind of information expressed. We further divide our definition of bias into two subcategories based on their cause: content bias and structure bias. These definitions draw on existing taxonomies of representational harms in NLP (Hovy & Spruit, 2016; Barocas et al., 2017), but are tailored to summarization.

We define **content bias** as bias towards the mention of a group in a text. In this type of bias, if group A is mentioned in the article, the summarizer will generate a biased summary. Changing the structure of the text will have no effect on how biased the summary is. We further divide content bias into five subcategories: under-representation, inclusion/exclusion, inaccuracy, sentiment bias, and framing bias, and explain each in Table 1.

On the other hand, we define **structure bias** as a bias as a result the structure of the text being summarized. In contrast to content bias, structure bias is invariant to the presence of information about different groups, but will change when document structure is modified. We define three structure bias subcategories: position bias, sentiment bias, and style bias, further explained in Table 1.

There are situations where the line between group and structure bias is not so clear cut. For example, if articles about group A are written in a way that a summarizer has a structure bias against, but articles about group B are written in a way that does not illicit this bias, the summarizer will show a consistent bias against group A. Though this is a result of a structure bias, it is also an example of content bias because the article structure and the group being discussed are dependent.Regardless of the cause, biased summaries have the potential for real, negative effects on marginalized groups.

## 3 OUR METHODS FOR MEASURING BIAS

Each of the types of bias defined in Table 1 represents a real problem in automatic summarization. However, we are not able to explore all of these biases in this paper, and some (i.e. framing bias) would be best explored using manual linguistic analysis than computational techniques. Instead, we aim to explore the content and structure bias of summarizers along three axes: underrepresentation,

---

[3]We recognize that it could be argued that a perfectly *unbiased* summarizer would not misrepresent groups at all, even if the original text does, this is a strong constraint to place on a summarization algorithm that does not have a concept of social bias.

Table 1: Potential ways in which a summarizer could show content and structure bias. Bias name appears on the left, bias category (content or structure) in the middle, and a description of the bias on the right. We consider all measures relative to the original article. We include an expanded table with examples of biased summaries in Appendix A

| Name | Category | Description |
|---|---|---|
| Underrepresentation | Content | Including less about one group than another in the summary. |
| Inclusion/exclusion | Content | In/excluding a detail only when a specific group is involved. |
| Inaccuracy | Content | Including information in a way that inaccurately reports information about a specific group (includes hallucination for abstractive models). |
| Sentiment bias | Content | Using more positive sentiment for one group over another. |
| Framing bias | Content | Using different sentence structure to frame a situation involving one group vs another. |
| Position bias | Structure | Selecting text from a specific position in the article |
| Sentiment Bias | Structure | Preferring information of positive or negative sentiment |
| Style Bias | Structure | Including sentences written in a specific style. For English this could include active/passive voice, first/third person, etc. |

position, and style. To this end, we design three experiments to measure these three different types of bias described in more detail in the following subsections.

**Data and Models** For our content bias analysis, we use a synthetic article corpus, the generation of which is explained in Appendix B. For all other experiments, we use the CNN/DailyMail corpus (See et al., 2017; Hermann et al., 2015). This corpus consists of 300k articles, written between 2007 and 2015, from CNN and the DailyMail. The corpus also contains reference summaries, generated from the bullet point highlights in each article. To generate summaries of these articles for our experiments, we use a four extractive and three abstractive summarizers described in Table 2.

Table 2: Summarizers used in the experiments for this paper. We choose a mixture of extractive summarizers, which generate summaries by picking relevant sentences from the article, and abstractive summarizers, which generate their own text. Azure and GPT-3 provide only API access, while all other models are downloaded from pre-trained checkpoints. All models except TextRank, which is fully unsupervised, use machine learning.

| Summarizer | Type |
|---|---|
| TextRank (Mihalcea & Tarau, 2004) | Extractive |
| PreSumm (Liu & Lapata, 2019) | Extractive |
| MatchSum (Zhong et al., 2020) | Extractive |
| Azure | Extractive |
| BART (Lewis et al., 2019) | Abstractive |
| PEGASUS (Zhang et al., 2020) | Abstractive |
| GPT-3 (Brown et al., 2020) | Abstractive |

## 3.1 CONTENT BIAS ANALYSIS

Given articles about different (combinations of) groups, do summarizers tend to include more information about some groups than others? For example, given articles that are half about men and half about women, their summaries should contain the same ratio of information. However, a summarizer with a bias against women may amplify information about men and supress information about women in its summaries. To explore this problem, we compare the amount of group information for each group in the original text to that in the summary. If information about a group is consistently reduced more or increased less than other groups, this indicates content bias in the summarizer.

In our analysis, we consider nine groups from three categories: men and women for gender; Black, White, Hispanic, and Asian for race/ethnicity; and Islam, Christianity, and Judaism for religion. To fully explore the effect of different combinations of groups, we use GPT-2 (Radford et al., 2019) to generate a synthetic corpus of articles designed to contain different ratios of information for group pairs from each category. We describe our methods for measuring the amount of group information in a text in the following subsections. Full methods for generating our synthetic corpus can be found in Appendix B

**Group Association Scores** In order to measure underrepresentation bias, we must first be able to measure how much information about a group is present. While this could be done using a keyword counting approach by creating lists of words corresponding to each group and quantifying the group information as the number of words from each list that appear in a text, this is likely to miss information as it is not feasible to specify every word related to a group. Instead, inspired by work measuring bias in static embeddings (Caliskan et al., 2017; Bolukbasi et al., 2016; Garg et al., 2018), we measure the group membership of each word in a text by generating its embedding using a word2vec model (Mikolov et al., 2013) and measuring its average cosine similarity to group word lists from Garg et al. (2018); Bertrand & Mullainathan (2004); Manzini et al. (2019).

After we have these scores, we aggregate the scores to obtain a text-wide measure of association with each group. We refer to this as a group association score, or $A$, and it is calculated as follows:

$$A(T, g) = \frac{1}{|T|} \sum_{t \in T, S(t,g) > \alpha_i} S(t, g) \tag{1}$$

Where $T$ is the text, $g$ is the group, $S(t, g)$ is the average cosine similarity between token $t$ and group $g$, and $\alpha_i$ is the cosine similarity threshold for group $i$ above which we consider words to have a strong similarity to the group $g$[4]. As it is highly unlikely that words belonging to some POS will have a true relation to a group, we use different POS filters to calculate these association scores depending on the group category. For gender, we calculate the scores only for pronouns and nouns and for race/ethnicity and religion we use nouns and adjectives.

For each group, we calculate the *representation score*, $R(T, g)$ as relative proximity to the group with the maximum difference association score in the text:

$$R(T, g) = \frac{A(T, g)}{max_y(A(T, y))}$$

This means that for a group with score $0.4$ over a set of texts, we would expect to see 40% the information about that group vs. the group each article is about. Unbiased summarizers should produce summaries with roughly the same proximity scores for each group as the article. If these scores decrease for a group after summarization, this indicates an underrepresentation bias toward that group. Likewise, if these scores *increase* for a group, this indicates a tendency to overrepresent this group. For a summary that does not show underrepresentation bias, we expect that representation scores for each group in the summary to be the same as those in the original article.

## 3.2 POSITION BIAS ANALYSIS

If information given at the end of an article is moved to the beginning, is it more likely to appear in the summary? As it is impossible to write about two different things in the same space, if there are multiple groups mentioned in an article, one must appear first. If summarizers show a bias for picking earlier or later information, this may also pose a fairness problem towards different groups.

It has been found that reference summaries and some automatically generated summaries for news articles show a bias for information that appears near the beginning of the articles Jung et al. (2019). However, it is not known if this pattern holds for more recent summarization algorithms, nor if the amount of position bias between human and automatic summarization is comparable. Our goal is to measure the amount of position bias in reference summaries to that shown by each summarizer

---

[4]For a description of how we calculate these thresholds, see the Appendix C.3.

to see if this existing bias is *amplified* through the use of automatic summarization. To do this, we compare the distribution of sentence locations between automatically generated summaries and reference summaries to measure any differences.

Because we examine abstractive and extractive summaries, we cannot measure where each summary sentence is from through exact matching. Instead, we create TF-IDF vectors of each sentence in the article and summary and compute the cosine similarity between each, marking article sentences with high similarities as being present in the summary.

### 3.3 Phrasing Bias Analysis

Given a paraphrased version of an article, do summaries still include the same information in their summaries? We assume that a summary should include the same information regardless of how it is phrased in the original text. However, we do not know if this is the case for automatic summarization. A summarizer that is sensitive to the phrasing of sentences may generate drastically different summaries of an article and a paraphrased version of that article. Fragility of this kind is highly undesirable. For one thing, it is highly unintuitive to human authors. If one article contains the exact same information as another, we expect that their summaries should also contain the same information, regardless of article phrasing. But additionally, sensitivity to phrasing could lead to amplified content bias. Imagine an author who consistently uses a certain phrase in sentences about one group and a summarizer that never chooses sentences with this phrasing. This summarizer would then unfairly exclude information about this group due to its sensitivity to phrasing.

We perform two paraphrasing experiments to measure the sensitivity of each summarizer to sentence phrasing: one targeted version and one random version. In the targeted version, we paraphrase the article sentences that originally appear in the generated summary. In the untargeted version, we paraphrase a random 10% of sentences in the article. We compare the sentencees that appear in the original summary to those that appear in the newly generated summaries. If the summarizers are sensitive to sentence phrasing, we should observe lots of changes in the sentences selected for the original summary and the summary after paraphrasing.

## 4 Results

The results of our experiments show that automatic summarization is sensitive to the structure of articles, as well as, to some degree, the groups discussed in the articles. Additionally, when compared to the reference summaries and the original articles, machine learning based summarization algorithms show a tendency of *exacerbating* the position bias shown by the reference summaries. Due to space constraints, we report only our results for gender. However, our results hold for other groups as well, and are presented in the appendix.

### 4.1 Content Bias Analysis

We summarize articles comprised of half sentences from articles about men and half about women. We find that all machine-learning based algorithms show a large sensitivity to the ordering of the information in the text, suggesting a position bias, which we explore further in Section 4.2. As shown in Fig. 1, almost all machine-learning based summarizers show a strong preference for sentences about the group that appears first in the article. Even though there are an equal number of sentences in each article about men and women, these summarizers show a tendency to *amplify* information that appears earlier in the article and *surpress* information that comes later. GPT-3 is the exception, showing an opposite, but much stronger tendency. It prioritizes information that appears later in while information appearing earlier in a biased article, this pattern king for ordering GPT-3 suppress information that begin with 90% sentences about men and with 10% about women, all machine learning based models except GPT-3 consistently maintain or increase the men's representation scores while decreasing those of women, indicating a preference for information about men. However, when this order is reversed, this pattern follows suit Fig. 2b. Despite the small ratio of sentences about women present in the summaries, it is still amplified if it occurs in the preferred location in the article.

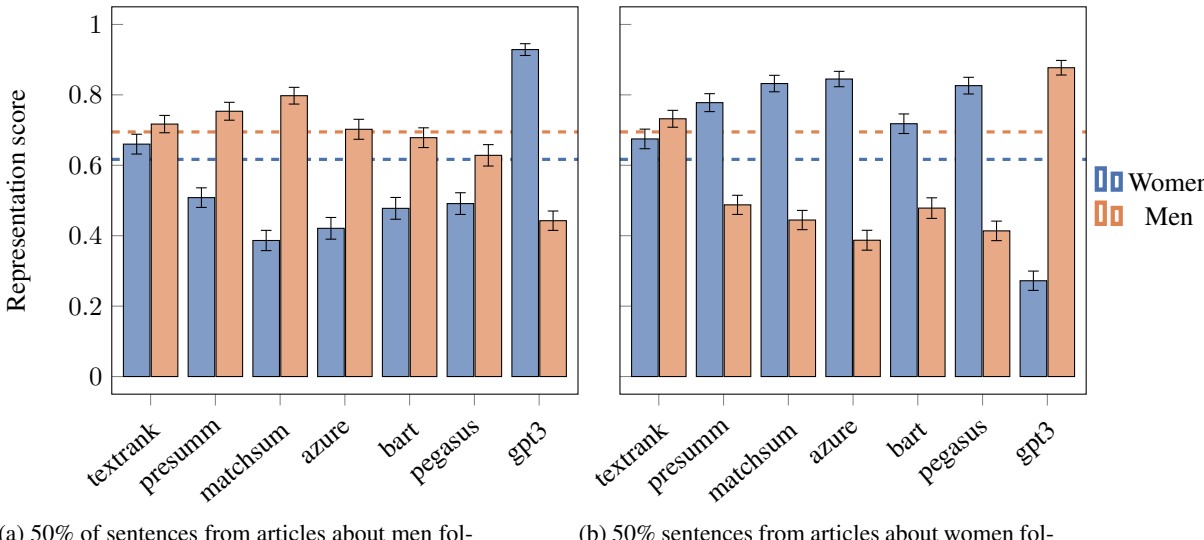

(a) 50% of sentences from articles about men followed by 50% from articles about women

(b) 50% sentences from articles about women followed by 50% from articles about men

Figure 1: Representation scores for men and women in summaries of balanced multi-group articles. Dashed lines are baseline representation scores of original articles, while bars scores of summaries. Error bars represent 95% confidence intervals.

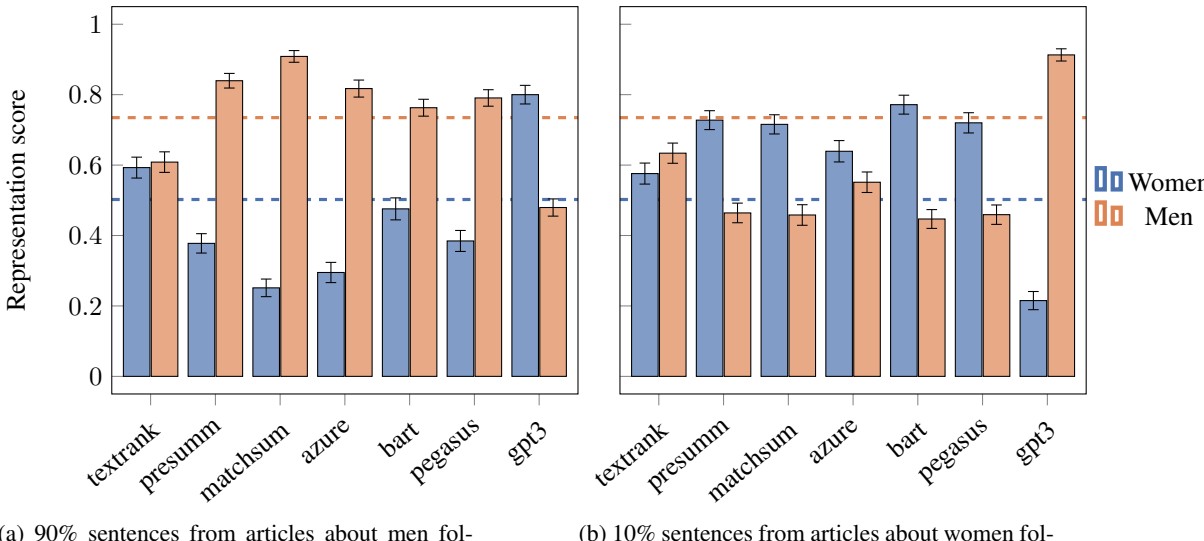

(a) 90% sentences from articles about men followed by 10% from articles about women

(b) 10% sentences from articles about women followed by 90% from articles about men

Figure 2: Representation scores for men and women in summaries of imbalanced multi-group articles. Dashed lines are baseline representation scores of original articles, while bars scores of summaries. Error bars represent 95% confidence intervals.

To see if all of the bias is due to location, we summarize articles about *only* men and women and measure if amplification/supression still occurs. As shown in Fig. 3b, we observe that for articles about men, no significant changes occur for either group. In contrast, in summaries of articles about women, the extractive summarizers maintain this pattern, but the abstractive summarizers show a tendency to *underrepresent* women and *overrepresent* men in their summaries. As seen Fig. 3a, this is particularly exacerbated when summary length is increased, with BART and GPT-3 making significant ($p < 0.05$) changes to the representation scores of men and women in summaries 50% the length of the original article.

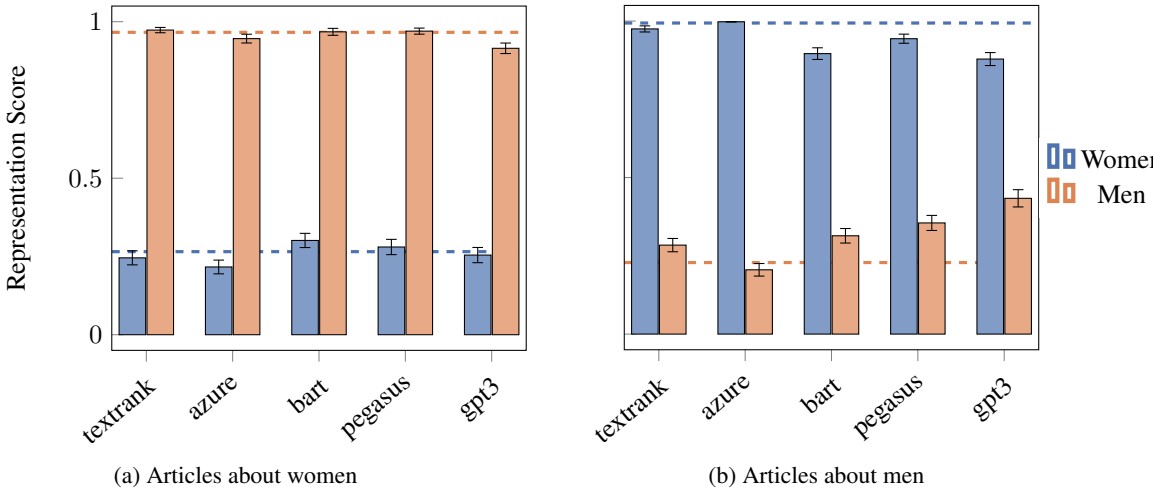

(a) Articles about women                    (b) Articles about men

Figure 3: Representation scores for men and women in summaries 50% the length of single-group articles. Dashed lines are baseline representation scores of original articles, while bars scores of summaries. Error bars represent 95% confidence intervals. Here, because we cannot modulate the length of generated summaries, MatchSum and Presumm are not included.

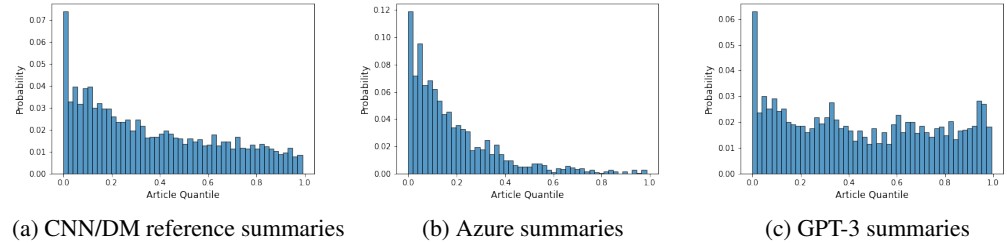

(a) CNN/DM reference summaries      (b) Azure summaries      (c) GPT-3 summaries

Figure 4: Percent of sentences chosen from each position in original articles from the CNN/DailyMail corpus by the reference summaries and by Azure and GPT-3.

Overall, these results show a pattern of unpredictability and fragility in machine learning based summarization algorithms, which seem to depend more on the ordering than the presence of information. While some ML based summarizers are more susceptible to this than others (e.g., GPT-3), all exhibit similar tendencies.

## 4.2    POSITION BIAS ANALYSIS

The pattern we observed in Section 4.1 for summaries on our synthetic corpus also holds for summaries of real articles from CNN/DailyMail. All summarizers except GPT-3 show a preference for sentences that appear close to the start of articles. This preference exists in the reference summaries as well, however, it is clearly *amplified* by MatchSum, PreSumm, and Azure, as shown in Fig. 4b[5] (all machine learning based abstractive models), which select sentences from the start of articles more often and almost never select sentences from the end of articles. In contrast, GPT-3 selects sentences from the start of articles *less* often, and selects sentences from the end at a significantly higher rate than the reference summaries Fig. 4c. These inconsistencies show marked differences between human and automatic text summarization, and, in combination with the content bias results shown in Section 4.1, suggest that machine learning based summarizers may be learning more about *where* to select information from and less about *what* information to select.

Our paraphrasing analysis shows that summarizers are sensitive to the structure of the articles they summarize. Specifically, however, they are sensitive to changes in the structure of sentences that

---

[5]Consult Appendix E for all figures

would have appeared in the original summary. As shown in Fig. 5a, when the sentences chosen for paraphrasing are those that appear in the summary of the un-paraphrased article, the likelihood that the summary will remain the same is quite low. However, when a similar amount of random sentences are paraphrased in the untargeted version, often the summary of the paraphrased article uses the same sentences as the un-paraphrased version, as Fig. 5b illustrates. This demonstrates that summarizers are highly sensitive to the paraphrasing of sentences that they rank as highly important, showing a structure bias.

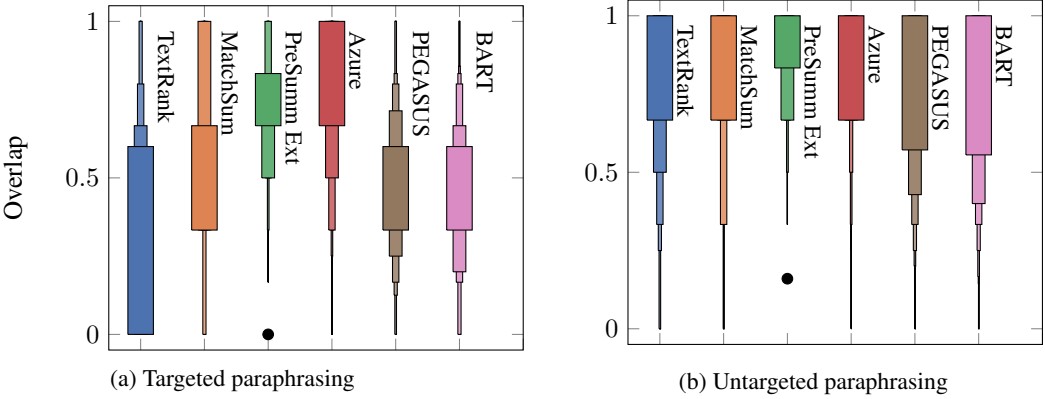

(a) Targeted paraphrasing           (b) Untargeted paraphrasing

Figure 5: The proportion of overlap in sentences chosen for summaries before and after paraphrasing for each of the 5 paraphrased versions of each article. While all are relatively unstable in the targeted version (Fig. 5a), in the untargeted version (Fig. 5b), the percent overlap is noticably higher.

## 5 LIMITATIONS

**Synthetic Data** Using news articles to study the content bias of summarizers comes with a number of challenges. First, news articles are not always about specific groups. Additionally, some combinations of groups show up very rarely, making it impossible to get data about them. Ideally, we would modify the amount of information present for each group to measure how much summarizers prioritize some groups over others. Using synthetic data gives us the flexibility to limit the groups present in an article or combine different amounts of information about specific groups to measure the impact of both group and amount of information on generated summaries.

**Types of Bias** We are limited to exploring types of bias that can be reliably detected using computational methods. This ignores many more subtle types of bias, such as misrepresentation, which are no less important for not being included in our analysis. This is not intended as an exhaustive study of the types of bias that can occur in automatic summaries. Rather, it is an initial exploration which we hope will inspire others to consider the bias summarization systems may introduce. Other types of bias, such as framing bias would be best explored using manual linguistic analysis, and this is a line of future research we strongly encourage.

## 6 RELATED WORK

To best understand our work, it is important to have an understanding of automatic summarization itself, as well as linguistic and NLP methods for measuring bias, and prior work analyzing various types of bias present in automatic summarization.

**Summarization Methods** There are two methods of automatic summarization: extractive and abstractive. In extractive summarization, an articles' sentences are ranked according to some algorithm and the top $k$ are chosen as the summary of the text. We study the extractive models TextRank (Mihalcea & Tarau, 2004), which is based on the PageRank algorithm, along with PreSumm (Liu & Lapata, 2019) and MatchSum (Zhong et al., 2020), which use embeddings and neural networks to score and select sentences for a summary.

In abstractive summarization, the summarization model generates every word of the summary itself rather than taking from the article directly (though some models, notably Pointer-generator (See et al., 2017), do have mechanisms to directly copy). As these models are generative, they generally rely on neural nets using typical encoder-decoder architecture. Examples of models that have been used for abstractive summarization include PreSummAbs (Liu & Lapata, 2019), the abstractive version of PreSumm, the language model BART (Lewis et al., 2019), and PEGASUS (Zhang et al., 2020), a language model designed by Google with summarization in mind. All of these models have obtained SOTA performance at some time or another for summarization on the CNN/DailyMail (See et al., 2017; Hermann et al., 2015) dataset.

**Linguistic and NLP measures for bias**   There have been various studies and measures of bias in NLP, notably in word, both quantifying the bias in embeddings (Bolukbasi et al., 2016; Caliskan et al., 2017) and using embeddings to measure the bias in human text (Garg et al., 2018). Closely related to our work, Dacon & Liu (2021) measure gender bias in news article abstracts using a variety of computational techniques, finding that they are overwhelmingly biased against women.

In linguistics, studies of bias generally use a combination of lexical analysis and discourse analysis to measure how members of different groups talk, how they are talked *about*, and often what stereotypes exist around both (Mendoza-Denton, 2014; Baker, 2012; Fairclough, 2013). Automated analysis restricts our ability to use techniques that rely on more qualitative analysis, but we draw inspiration from them in our definitions of bias and our experiments.

**Properties of Automatic Summarization**   Dash et al. (2019) explore a related topic to this paper, measuring fairness in automatically generated tweet collections by user demographic, finding that existing extractive methods exhibit bias toward tweets written by different groups, dependent on the corpus used. Keswani & Celis (2021) perform a similar line of work, exploring dialect diversity in Twitter summaries and finding that summarizers prefer tweets in Standard American English (SAE) over other dialects.

Jung et al. (2019) find that, for news articles, human summaries tend to select information from near the start of articles. Additionally, they show that various summarization algorithms place great importance on the position of information. Lee et al. (2022) measure and attempt to mitigate political framing bias in summaries of news articles by encouraging the generation of summaries that synthesize political views from a variety of sources. This relates to work from Sharevski et al. (2021) who show it's possible to adversarially bias a summarizers toward writing in a particular political writing style. Finally, Jørgensen & Søgaard (2021) explore possible bias in summary evaluation by conducting a pilot study exploring how user demographic relates to perception of summaries. Their results show that generated summaries are most strongly preferred by older White men, while other demographics rank them more poorly.

# 7 CONCLUSION

In this paper we have shown that **machine learning based summarization algorithms demonstrate bias toward under-represented groups and are highly sensitive to article structure**. What groups are mentioned, where information appears in the article, and how important sentences are phrased all have large effects on the generated summaries.

All of these exhibited biases have potential to cause harm. Readers rely on summaries to accurately represent a longer text, and if a summarizer systematically introduces position, sentiment, or content bias this risks both mis-representing the original text and biasing the reader. Although this bias is not intentionally designed into these algorithms, it should be tested and corrected for in future summarization algorithms. Especially as automatic summarization is marketed more and more as both a commercial and personal-use tool, it is important to measure and correct the bias in these systems *before* deploying them.

While a summarizer may obtain a high ROUGE score, this does not necessarily mean that the summaries produced will fairly reflect the articles they represent. They may be biased in how they refer to certain groups, in the sentiment they use, or in terms of the article structure they favor. We recommend that more bias evaluation methods, like the ones used in this paper, be used when evaluating new summarizers to avoid creating unintentionally biased summarizers in the future.

## 8 ETHICAL CONSIDERATIONS

Any study of bias inherently carries ethical considerations. A summary that amplifies bias already present in the original article or generates new bias has potential to cause harm. Especially as automatic summarization is marketed more and more as both a commercial and personal-use tool, it is important to explore the problems this could create. Sharevski et al. (2021) have shown that it is possible to bias a summary to a specific political ideology, and our work has shown that even without modification, different summarizers treat groups differently and are sensitive to changes in article format. This raises concerns about summarization that should be kept in mind when designing and evaluating models in the future.

In our analysis we consider only one corpus, which contains writing with a relatively uniform writing style, and is written in Standard American English. This does not allow us to explore bias in language style or dialect, which is another possible axis of bias in summarizers.

Our motivation for using embedding similarity scores to measure group membership is to allow for flexibility that wordlists cannot provide. As word embeddings have been shown to encode factual information, using these scores is a good way to achieve this. However, these scores have also been shown to encode human biases and stereotypes, which we found to be true in our analysis as well. For example "terror" was the most associated word with the category "Islam" in some examples. While in some cases articles may actually be using these words with stereotypical associations to refer to groups, this is not always the case, and could introduce some amount of error. Based on our observations, this risk is small, but does represent a possible limitation.

## 9 REPRODUCIBILITY

We provide all code used to perform our analysis, along with the GPT-2 generated corpus. Full details for both how we generated this synthetic corpus, and how we performed our analyses are provided in our appendix. All other data and models are publicly available and we do not modify them. Though some of the models we use exhibit randomness in the summaries they generate, we have run sufficient experiments that we are confident our results could be reproduced.

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

## A  TYPES AND EXAMPLES OF CONTENT BIAS

Table 3: Potential ways in which a summarizer could be biased with respect to a group. Highlighted original text is text *not* selected that highlights the bias.

| Category | Description | Original Text | Summary |
|---|---|---|---|

Table 3: Potential ways in which a summarizer could be biased with respect to a group. Highlighted original text is text *not* selected that highlights the bias. (Continued)

| | | | |
|---|---|---|---|
| Over-representation | Including more about one group than another relative to the ratio in the original text | The women's team scored very high in the competition. ==They practiced for many months to get here and it paid off. Their scores set a new record.== ... The men's team did pretty well too. They beat a few other teams. | The women's team scored very high in the competition. The men's team did pretty well too. They beat a few other teams. |
| Inclusion/exclusion | In/excluding a detail only when a specific group is involved | Both the women's and men's figure skating teams are headed to the Beijing olympics. They will compete throughout February. Both are expected to win lots of medals. ... We've gotten a chance to see their costumes early. The women's costumes are very pretty. ==The men's costumes are also very good.== | Both the women's and men's figure skating teams are headed to the Beijing olympics. Both are expected to win lots of medals. The women's costumes were very pretty. |
| Inaccuracy | Including information in a way that inaccurately reports information or hallucinating inaccurate information (for abstractive models) about a specific group | What to do if someone's having a seizure: Move them to a safe position. Roll them to their side. .... It's also important to know what not do to do. ==Here's a list of things not to do if someone's having a seizure:== Hold them down. Put something in their mouth. | What to do if someone's having a seizure: Hold them down. Put something in their mouth. |

Table 3: Potential ways in which a summarizer could be biased with respect to a group. Highlighted original text is text *not* selected that highlights the bias. (Continued)

| | | | |
|---|---|---|---|
| Sentiment bias | Using more positive sentiment sentences for one group than another | Two common pet choices are dogs and cats. One may be better than the other depending on your lifestyle. ==Dogs are very loving and make great pets.== Dogs require a lot of attention and care and need to be walked every day. Cats can be very affectionate and are great pets. ==Cleaning up after cats is also a lot of work.== | Two common pet choices are dogs and cats. Dogs require a lot of attention and care and need to be walked every day. Cats can be very affectionate and are great pets. |
| Framing bias | Using different sentence structure to frame a situation when one group is involved vs another | ==A police officer fatally shot a man at 8pm last night.== ... An officer's gun discharged, killing the man and injuring another last night. | An officer's gun discharged, killing the man and injuring another last night. |

## B    Synthetic Corpus Generation

We generate a synthetic corpus of articles using GPT-2 (Radford et al., 2019) and prompts from the BOLD dataset (Dhamala et al., 2021) for each group[6]. We refer to these articles as *single-group* articles, as they are prompted to generate articles about each group. We also create *multi-group* articles by sampling different amounts of contiguous sentences from single-group articles and combining them to form a new article with information about two groups. We generate multi-group articles for each combination of two groups within each of our three group categories, constructing one version with the first group at the beginning of the article, and another with this group at the end of the article (using the same sentences). This allows us to explore the effect of different group combinations and amounts of information while controlling for the importance of the order information appears in.

We generate our text using the version of GPT-2 available on the HuggingFace Hub (Wolf et al., 2019) as of September 2022 using the following parameters:

- `min_length=400`
- `max_length=1024`
- `top_k=40`
- `top_p=0.95`
- `no_repeat_ngram=3`

Additionally, for each group, we set `bad_words_ids` to the union of all other groups' keyword lists, which we find helps to keep articles "on track" about the group they're supposed to be about. Without this, for example, many articles about women talk mostly about men and vice versa.

## C    Content Bias Analysis Details

### C.1    Word Lists

We use lists of representative keywords for each of the nine groups we study. These lists are used for both filtering articles and the group analysis we perform using word embeddings. The lists for the male, female, Hispanic, Asian, Islam, and Christianity groups are taken from Garg et al. (2018) with some minor changes made to better suit the corpus. For example, we remove the word "cross" from the Christian word list as it is often used outside the context of Christianity in our corpus. We use the same lists as Caliskan et al. (2017) for the White and Black group lists (originally from Bertrand & Mullainathan (2004)), which uses first names rather than last names as done for the other racial groups. This is because, as Garg et al. (2018) describe, there is a very large overlap in the typical last names of white and Black people in the US, owing in large part to the country's history of slavery. For the Jewish group, we use the list used by Manzini et al. (2019).

### C.2    Group Similarity Scores

We generate embeddings using a word2vec model (Mikolov et al., 2013) pre-trained on the Google News corpus. Before measuring similarity, we tokenize and POS tag each document with NLTK (Bird et al., 2009) and the universal tagset (Petrov et al., 2012) (this allows us to filter words by POS, which we explain further in Section 3.1). Then, we measure the group similarity for each token embedding, $t$ in the text as follows:

$$S_i(t) = \frac{1}{|G_i|} \sum_{g \in G_i} s(t,g) \tag{2}$$

Where $G_i$ is $i$th set of group keywords, and $s(t,g)$ is the cosine similarity between the two token embeddings.

We collect these scores for each word in both the original article and its summary and aggregate them into text association scores, described below.

---

[6]For detailed information on generation parameters, see the appendix.

We observed that, though these scores serve as a good indicator of the groups present in an article, they can be quite noisy for words with low scores. This noise comes as a result of many words in each article having a baseline similarity to people (something all the groups have in common) and results in the majority of words in an article receiving quite low scores, while the minority, most indicative of group presence in the text, receive very high scores. Therefore we focus on only the high scoring words to determine group membership and discard lower scores as uninformative.

### C.3 THRESHOLDS

The thresholds used to determine high scoring words were calculated using similarities of each group list to the other group lists. For each group, we take the median similarity of the most similar alternate group to be the threshold above which we consider words important. In practice, we find that this reduces the number of words that really belong to one group that are given falsely high association scores for other groups.

### C.4 ARTICLE FILTERING

As an additional filter, we restrict our analysis to articles that contain at least one word from at least one of our group keyword sets. This is to ensure that at least one group is truly present in the article, and does not significantly reduce the number of articles we are examining (the original CNN_DM dataset is 311,971 articles, and with this filter, we analyze 296,460 articles). We recognize that bias could still be present if a group is not explicitly mentioned; our group keywords are not all encompassing and there are many implicit ways to signal group information (e.g., dog-whistling Haney-López (2014)). However, manual linguistic analysis is far more suited for exploring this type of bias, and we find that this filtering makes our analysis more reliable.

## D PARAPHRASING ANALYSIS DETAILS

We use the pre-trained PEGASUS paraphraser[7] from HuggingFace (Wolf et al., 2019) for these experiments and run the analysis on the test split of CNN/DailyMail (˜11k articles). We create five versions of each paraphrased article in order to measure the effect of different paraphrases on which sentences are selected for an article.

## E ADDITIONAL LOCATION ANALYSIS PLOTS

As discussed in the main paper, we find that Presumm, MatchSum, and Azure show a strong tendency to select sentences that occur early in articles—stronger than that demonstrated by the reference summaries. In contrast, we find that GPT-3 prefers information that comes at *either* the start or the end, with a much stronger preference for information at the end than shown by reference summaries.

Note that Presumm and MatchSum have relatively short length limits of 512 tokens, which means they are unable to select sentences from the end of long articles. This impacts these figures, and is the reason for the dropoff in frequency between 0.4 and 0.6 in their plots. However, if this were the only effect, we would expect that there would be more sentences selected between 0.1 and 0.4 for both summarizers, and not the sharp spike in sentences taken from the first 2% of the article, as we see with MatchSum and Presumm.

---

[7]https://huggingface.co/tuner007/pegasus_paraphrase

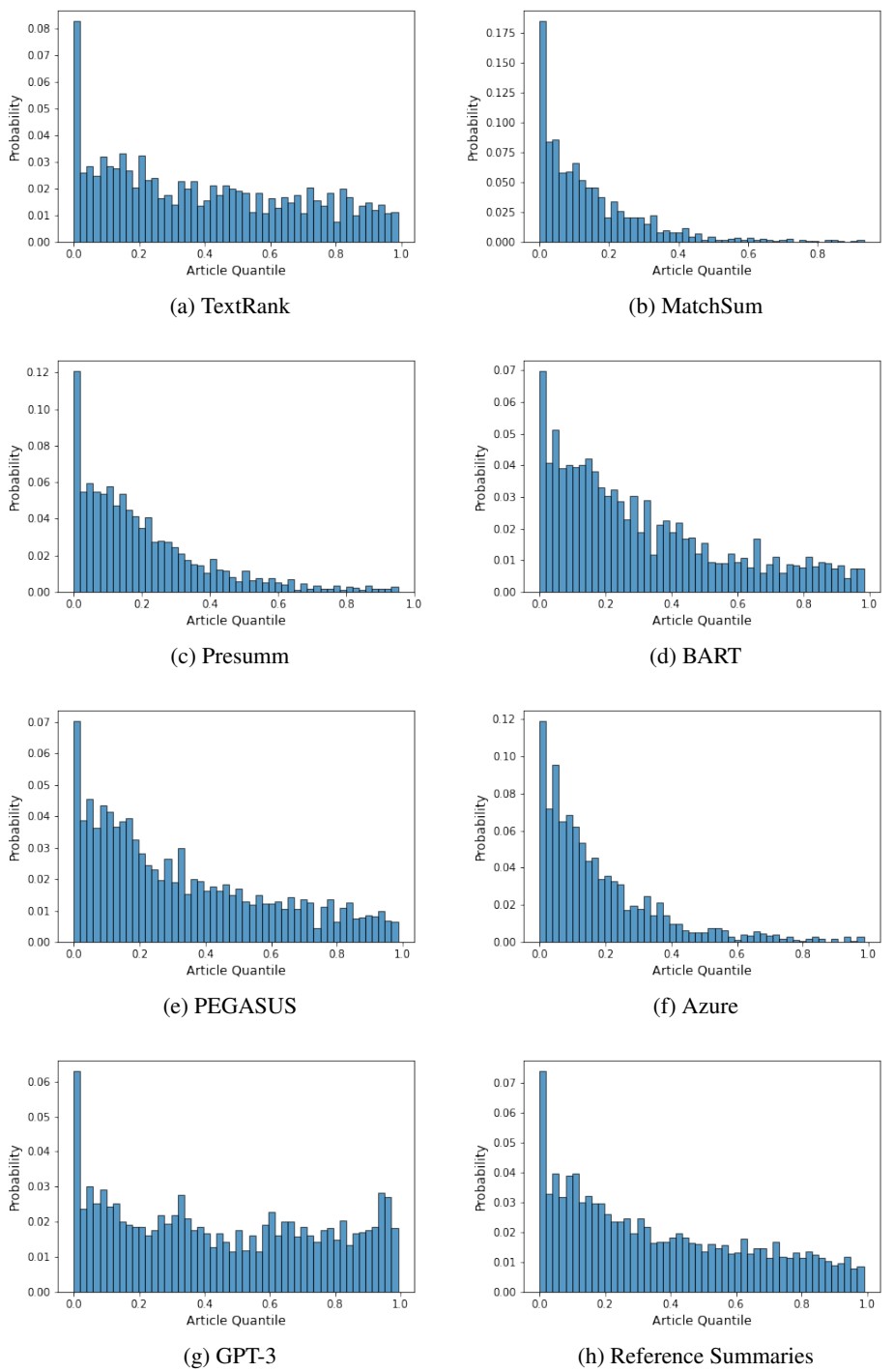

(a) TextRank

(b) MatchSum

(c) Presumm

(d) BART

(e) PEGASUS

(f) Azure

(g) GPT-3

(h) Reference Summaries

## E.1 GENDER

### E.1.1 RECURSIVE SUMMARIZATION

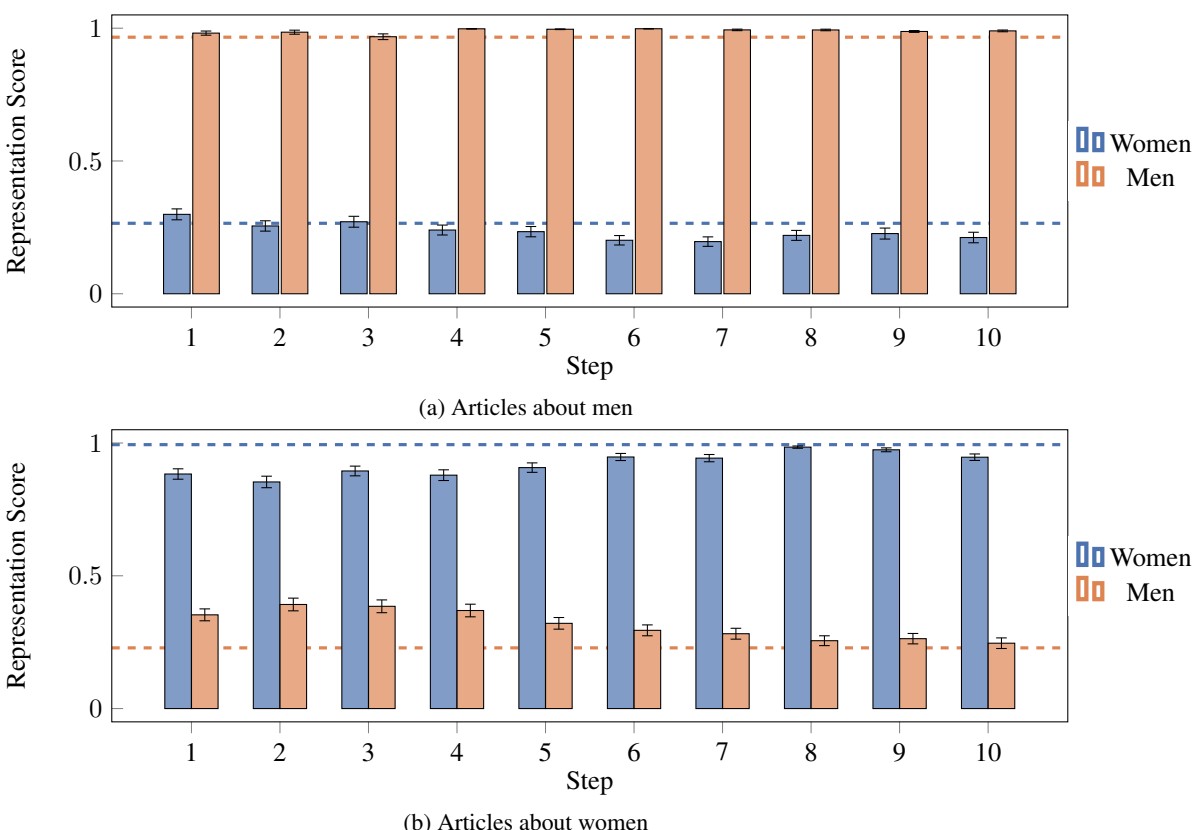

(a) Articles about men

(b) Articles about women

Figure 7: Ratio of information present about each group relative to the highest group information present in the article. A score of 1 for a group and summarizer means that a reader could expect to see that group with the most information in every summary we generated using that summarizer. A score of 0.4 means the reader could expect to see 40% of the information about that group vs. the group with the most information in each summary.

Here we observe similar patterns to that seen in single group summarization. Abstractive summarizers show a tendency to first gain, then lose information about men for articles about women, and lose and then gain information about women. However, for articles about men, the amount of information about both groups stays relatively constant across all steps of summarization.

## E.2 RELIGION

For single group articles about different religions, we observe that while articles about Islam generally preserve the baseline amount of information from the article, summaries of articles about Judaism and Christianity tend to lose information about their respective groups (shown in Fig. 8). However, unlike in the gender case, the representation scores of other groups do not tend to increase, suggesting that the overall information about religion in these summaries is being decreased, rather than the emphasis being shifted to another religion.

Similar patterns to what we see for gender also hold for religion. In multi-group articles about Judaism and Christianity, regardless of which group has more information in the original article, the prioritized group in the summary has more to do with which information is presented first. As shown in Figs. 9 and 10, all summarizers but GPT-3 show a preference for text that appears earlier in the article.

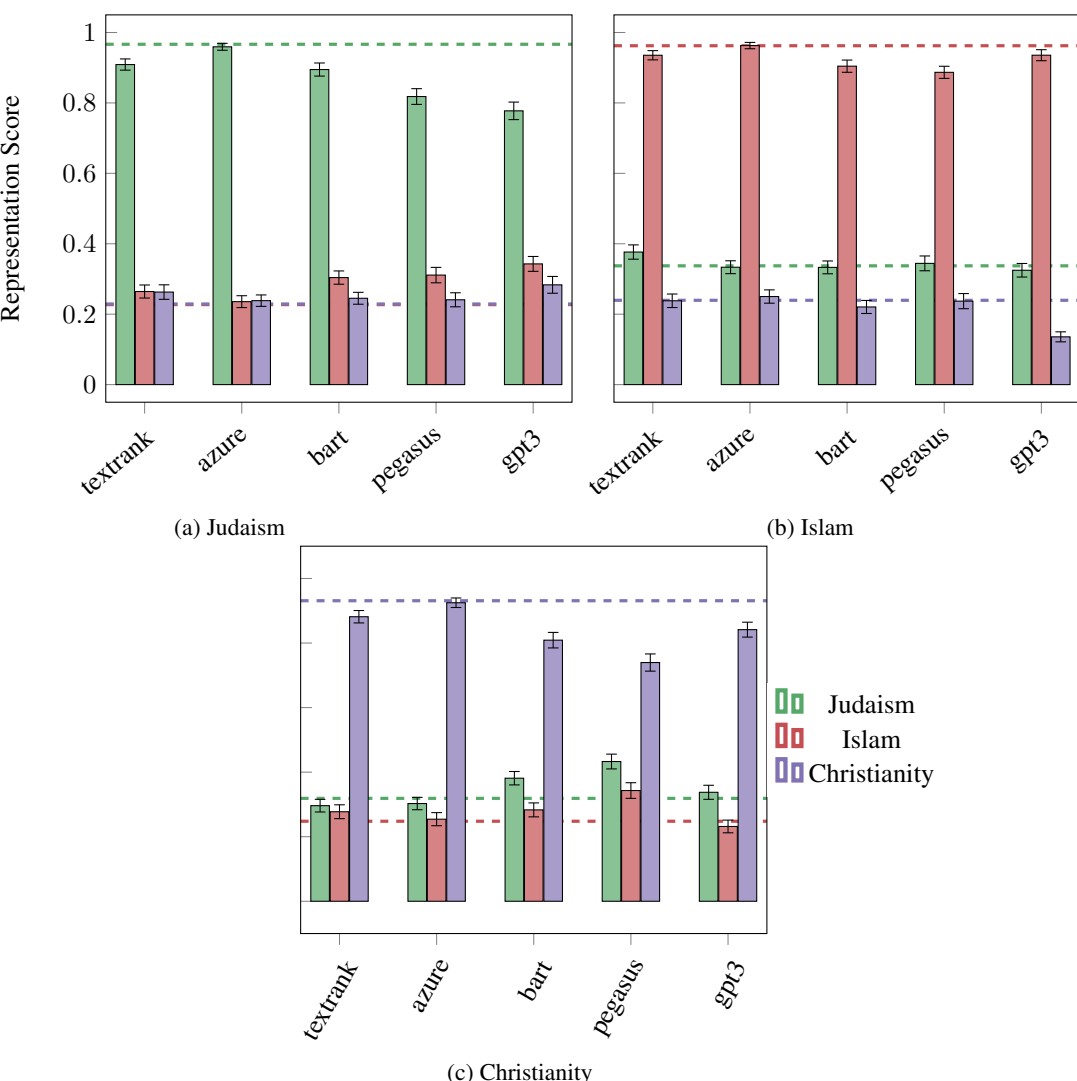

Figure 8: Representation scores for summaries 50% the length of articles about single groups. Dashed lines are baseline representation scores of original articles, while bars scores of summaries. Error bars are the 95% confidence interval for each score.

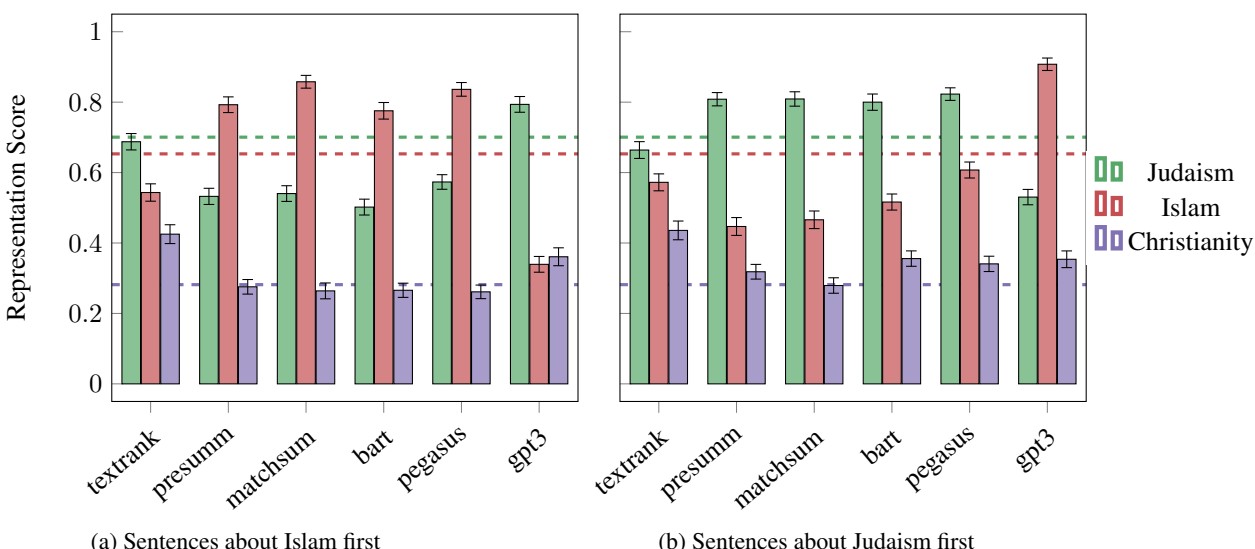

(a) Sentences about Islam first

(b) Sentences about Judaism first

Figure 9: Representation scores for summaries 10% the length of articles. Articles are comprised of 50% sentences about Judaism and Islam. Dashed lines are baseline representation scores of original articles, while bars scores of summaries. Error bars are the 95% confidence interval for each score.

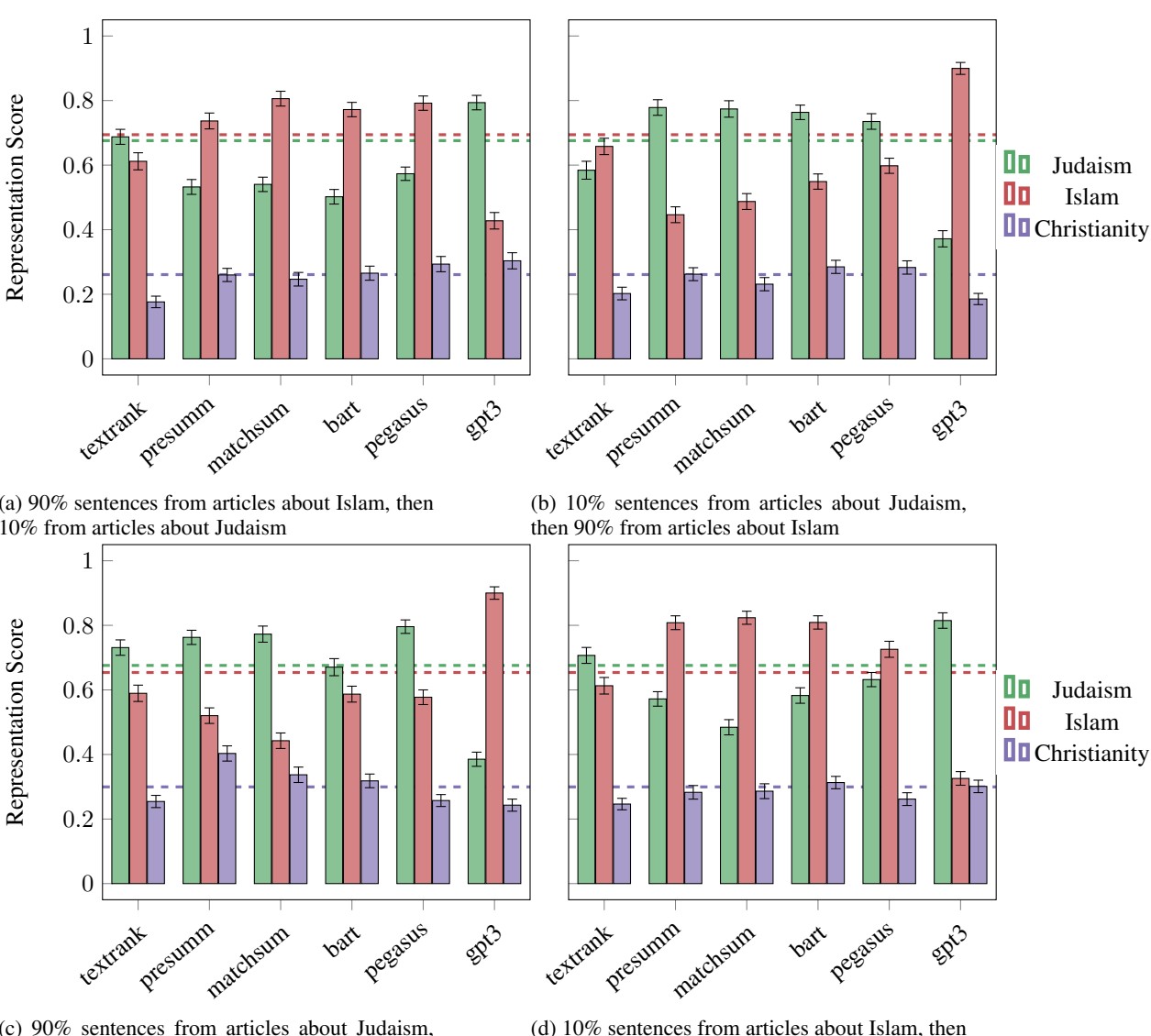

(a) 90% sentences from articles about Islam, then 10% from articles about Judaism

(b) 10% sentences from articles about Judaism, then 90% from articles about Islam

(c) 90% sentences from articles about Judaism, then 10% from articles about Islam

(d) 10% sentences from articles about Islam, then 90% from articles about Judaism

Figure 10: Representation scores for summaries 10% the length of unbalanced articles about Judaism and Islam. Dashed lines are baseline representation scores of original articles, while bars scores of summaries. Error bars are the 95% confidence interval for each score.

### E.3   RACE

For race, the text association scores have several problems. While they accurately indicate the presence of information for both the Hispanic and Asian groups, for the Black and White groups, we find that the scores are not good indicators of what information is there. Black scores are consistently high for articles about *all* groups, and especially correlate with White scores. This same problem has been observed in other work that uses word lists based on names for race (Garg et al., 2018), and is not surprising. However, it does mean that we cannot report definitive results for race using our scores.

