# OpenReview forum: "How (Un)Fair is Text Summarization?"
_ICLR.cc/2023/Conference — Submitted to ICLR 2023_

### Official Review · Reviewer_KeGL · 2022-10-17

**Confidence:** 4
**Correctness:** 3
**Technical Novelty And Significance:** 4
**Empirical Novelty And Significance:** 3
**Recommendation:** 5

**Clarity, Quality, Novelty And Reproducibility:**

Overall, the paper is straightfoward to follow, the novelty is great and the analysis is interesting and reproducible with reasonable \
effort.

I think the paper would be much stronger if the metrics were validated using some human analysis. For example, can you select some gro\
ups and show that humans find the representation score appropriate? Moreover, can you select 10-100 samples that are both analyzed by \
humans and the automated scores and show that the scores match the human evaluation?

There have been quite a bit of evidence that bias metrics at the embedding level do not follow extrinsic measures of bias and such hum\
an validation would make the results more convincing.
On the Intrinsic and Extrinsic Fairness Evaluation Metrics for Contextualized Language Representations
Intrinsic Bias Metrics Do Not Correlate with Application Bias

Is there a reason why word2vec was used to create the representation scores (and not something newer, some LM-embeddings)?

I have a few suggestions for improvement:

* In the intro, when describing types of biases, show examples, much easier to follow with examples (and the reader is motivated to st\
ay engaged). Expand Table 1 with examples.

* List challenges for bias estimation in text summaries.

* Include a validation for metrics used as explained above

* Fig 1: Do a and b differ in the order in which the groups appear in the text (i.e., a men first, women second and b women first, ben\
 second)?

* I don't seem to find Footnote 4.

* Fig 7, no legend so not sure what the bar/colors/lines mean


UPDATE: Thank you for your response, I will maintain my scores.

**Strength And Weaknesses:**

Strengths:
- first study that I am aware of that looks at bias in text summarization
- discussion on types of biases possible in text summarization
- an intial study of bias in text summarizers

Weaknesses:
- I'm not convinced whether the proxies/metrics for measuring content bias are accurate
- Lack of some human validation

**Summary Of The Paper:**

This paper describes the typeAs of biases that can appear in automatic
text summarization and presents an initial study of such biases using
two corpuses: a synthetically generated one and the CNN/Daily mail
corpus.

The paper selects a few types of biases that can be analyzed
automatically and proposes metrics (or proxies) for assessing biases,
such as content bias wrt representation and structure biases.

Across several models analyzed (both extractive and abstractive),
there seem to be a general tendency to minimize the representation of
a group over the other. Similarly, most summarizers (with the
exception of GPT-3) seem to prefer sentences appearing at the
beginning of the text over sentences at the end of the text.

**Summary Of The Review:**

The paper is a first to analyze bias in automatic text summaries and
it would improve in quality if a human validation of metrics used and
results obtained would be included. I strongly encourate the authors
to include such a validation.

---

> ### Author Response · Authors · 2022-11-18
> **Author Response**
>
> Thank you for your comments.
>
> > I think the paper would be much stronger if the metrics were validated using some human analysis. For example, can you select some groups and show that humans find the representation score appropriate? Moreover, can you select 10-100 samples that are both analyzed by humans and the automated scores and show that the scores match the human evaluation?
>
> We designed our scores in such a way that they should correlate with human judgement. Static embeddings have been shown to correlate with real world biases and factual information (TODO: cite), and our use of similarity to establish group membership is quite similar to this work. Our final association scores are also similar to the type of scores we would be able to elicit from humans; rather than reporting absolute amount of information about a group, we compare distance from the group with the most information in an article. We do this because even asking people to score how much information about a group is present in an article will almost certainly lead to inconsistent results (both between different people and within one person's scores), and we believe this makes our scores easier to interpret. Nevertheless, verifying our scores using human judgement is a good suggestion.
>
> > There have been quite a bit of evidence that bias metrics at the embedding level do not follow extrinsic measures of bias and such hum an validation would make the results more convincing. On the Intrinsic and Extrinsic Fairness Evaluation Metrics for Contextualized Language Representations Intrinsic Bias Metrics Do Not Correlate with Application Bias.
>
> This is true, however we do not believe this applies to our definition of underrepresentation bias. Here bias is defined as a difference in the amount of information included about a group between the article and the summary. We use word embeddings to measure the amount of information about a group, which is something that word embeddings *have* been used for in other work (word embeddings quantify, etc)
>
> > Is there a reason why word2vec was used to create the representation scores (and not something newer, some LM-embeddings)?
>
> We use w2v precisely because of this and because contextual embeddings seem excessive for  our group membership heuristic. Computing LM embeddings comes at a higher cost and we felt the benefit it would add (possibly clarifying ambiguous cases) was unnecessary for our goal of having an easy to compute heuristic for group membership.
>
> > In the intro, when describing types of biases, show examples, much easier to follow with examples (and the reader is motivated to stay engaged). Expand Table 1 with examples.
>
> Thank you for this suggestion. We agree that examples are easier to follow, but are somewhat limited by space for including them in the main paper. We do include examples of each type of bias in the table in the appendix. We have added a pointer to this in our revision.
>
> > Fig 1: Do a and b differ in the order in which the groups appear in the text (i.e., a men first, women second and b women first, ben second)?
>
> Yes, this is exactly right. We have added additional clarification of this in our revision.

---

### Official Review · Reviewer_4JiR · 2022-10-24

**Confidence:** 4
**Correctness:** 2
**Technical Novelty And Significance:** 2
**Empirical Novelty And Significance:** 2
**Recommendation:** 5

**Clarity, Quality, Novelty And Reproducibility:**

The paper is not easy to understand, details are sparse and it is hard to judge the validity of the experiments without having a complete view of the setup. Some of the insights are novel but I can't validate their correctness as there are not enough details in the paper to help me understand what is going on in each experiment. Reproducibility is also difficult for the same reason. The Supplementary material is a zip file of zero bytes that does not expand so I can't run any code either.

**Strength And Weaknesses:**

## Strengths:
- Paper is well motivated. The scope of text summarization methods in practice points to a need to pay much more attention to this question.
- Some of the results offer interesting insights. Such as results in Figure 2, as it shows model reliance on what comes early in an article. But there are also weaknesses in this analysis as I share below.

## Weaknesses:
- The exposition is confusing, several details are missing and assumptions are made without declaration. There is no qualitative analysis to help the reader better follow how the authors are changing structure or content, or what a synthetic summary looks like, or what does it mean to be an article about _men only_ but still have sentences about _women_ in them (Figure 3) or whether the authors' several perturbation strategies aren't altering valuable information?
- The paper relies on the premise that a certain way of choosing a summary (say weighing what comes earlier in an article more), given an article is wrong. But it does not show what the alternative would be and whether that alternative is indeed more preferable to an end user. It is hard to say that given two different texts to an end user (since these are different as they've been reordered), they would somehow still write the same summary for both articles. There is also no end user analysis showing that these _biased_ summaries from a model that rely more on what comes earlier in the article are offering less value to the end users than what a counterfactual _unbiased_ summary would offer. Experiments in Figure 2 come close to this point but it does not come clearly. Summarization is not an objective task, so the gold standard is to show resulting summaries to humans and see whether they prefer one model or the other. But that's hard as the authors don't characterize what such an _unbiased_ summary would be?
- The paper appears rushed. Some examples: Figure 7 has no legend, Table 3's caption says some text is in red and other in blue, but the only color there is yellow. On Page 3, its said that the generation details for the synthetic corpus are in Section 5, on Page 4 its said that they are in Appendix B, but it seems they are in neither. Appendix B comes close to sharing some information but after reading it, I could still not tell how someone could replicate this synthetic data generation because details are sparse. Also, a footnote in this paragraph says that more details are present in the Appendix?
- Several claims have been presented as facts but no citations or data are offered to support them. Some examples:
  - Introduction: "Readers, however, expect that summaries faithfully represent articles." First, what do you mean by faithful in this context? Second, if this is the case, please add at least one behavioral science citation to this.
  - Footnote 3: What notion of fairness does this relate to?
  - Section 3.3: "While we could expect a human to generate a summary..." What is the evidence to support this expectation? Is there any data, any pilot study you did, any previous work that looks at this?
  - Section 5: "We conduct manual analysis..." No details are shared on this analysis and no results are presented.
- I have read the experiments and results section several times, yet I can't tell whether I would be able to replicate the setup and reproduce these results given how sparse the details are.
  - There are error bars in the plots. What do these represent?

**Summary Of The Paper:**

This paper looks at the problem of bias in automatic summarization algorithms. The authors design a method to quantitatively measure the extent to which an article's attributes, such as structure, paraphrasing, and article content influence _bias_ in generated summaries. They also study the causes of _bias_ by varying the influential factors, including the summarization parameters and distribution of input documents. The findings indicate that machine learning based summarization algorithms can introduce position or content bias, which can lead to misrepresentation of the original text and/or biasing the reader. The authors recommend that more bias evaluation methods be used when evaluating new summarizers to avoid creating unintentionally biased summarizers in the future.

**Summary Of The Review:**

The paper is well motivated, but the exposition is confusing and several details are missing. The authors rely on the premise that a certain way of choosing a summary is wrong, but they do not show what the alternative would be practically and whether that is indeed more preferable to an end user. The paper appears rushed, with some claims presented as facts without any supporting data or citations. It is also difficult to tell how one could replicate the experimental setup and reproduce the results given the sparse details.

---

> ### Author Response · Authors · 2022-11-18
> **Author Response**
>
> Thank you for your comments and suggestions.
>
> > What does it mean to be an article about men only but still have sentences about women in them (Figure 3)
>
> It's not possible to prevent GPT-2 from generating all words associated with a group, as this would require having a pre-defined list of such words. When generating articles about a group, we prevent GPT-2 from generating words from the keyword lists of other groups in that category (e.g., for men, it's prevented from generating the words about women we use to measure group similarity). However, this does not prevent the model from generating other words associated with other groups—especially names. This results in a small amount of information from other groups also being present in single-group articles.
>
> > No counterfactual examples of what an unbiased summary would be, and no end user study verifying assumptions about one thing being more biased than another (e.g. x is more biased than y)
>
> This can be hard to define, since as you say, summarization is a subjective task. We are able to provide examples of biased summaries under our definitions and do this in our appendix. A user study would be a good followup, however it is also difficult because of the subjectivity of both summarization and bias. Our metric should correlate well with judgement for bias, but not necessarily with judgement for usefulness or "similarity to human summaries". This makes an assumption that unbiased summaries are more similar to human summaries or more useful, which we do not do. This is the goal of utility evaluation metrics, not bias metrics.
>
> > The paper appears rushed.
>
> Thank you for pointing out these points. These have been fixed/clarified in our revision.
>
> > Introduction: "Readers, however, expect that summaries faithfully represent articles." First, what do you mean by faithful in this context?
>
> In our context, we assume that readers will be reading summaries instead of articles. These summaries, then should report the same salient information as the article (minus some lost to compression), and in the same manner.
>
> > Footnote 3: What notion of fairness does this relate to?
>
> This is a clarification on what our notion of fairness is. A more general definition of fairness may argue that models should not output any biased/toxic content towards any group, regardless of the original text. Our definition is somewhat relaxed in that it allows the summary to exhibit exactly the same level of social bias as the original text, so long as it isn't amplified.
>
> > There are error bars in the plots. What do these represent?
>
> Error bars represent 95% confidence interval. We have revised our figure captions to include this.
>
> > The Supplementary material is a zip file of zero bytes that does not expand so I can't run any code either.
>
> Thank you for pointing this out. There seems to have been some issue with the original upload of this. We have re-uploaded this data and verified that it can be downloaded properly.

---

### Official Review · Reviewer_45gx · 2022-10-24

**Confidence:** 3
**Correctness:** 3
**Technical Novelty And Significance:** 1
**Empirical Novelty And Significance:** 4
**Recommendation:** 3

**Clarity, Quality, Novelty And Reproducibility:**

Clarity
The paper is not very clear, there are a lot of loopholes that need to be fixed mainly concerning the explanation of the author’s ideas and implementation.
Quality
The quality needs improvement in terms of considering more experiments.
Novelty
The work is novel.
Reproducibility
Yes it seems reproducible as they have provided the codes and datasets.


**Strength And Weaknesses:**

Strengths
It addresses a significant problem largely ignored by the Automatic Text Summarisation Community of Bias Analysis.
It points out potential risks due to the ignorance of Bias analysis in Text Summarisation.
Defines different types of biases in Automatic Text Summarisation.
Good initial exploration to push the research in this direction.
Weaknesses
The paper does not compare the results of automatic summarizers with human curated summarization.
Only used single embedding word2vec (word embeddings can themselves be baised) and only considered one dataset CNN/Dailymail, without explaining any rationale behind the selection. To add value, authors should have explored other text summarization datasets with different writing styles to really bring out the different style biases rather than just paraphrasing.
The authors only considered a few models, ignoring sota models on CNN/Dailymail dataset. It may have been possible that the SOTA model behaves differently.
The paper lacks any association/description between the metrics used for bias in this paper v/s different text summarisation metrics. It would have been nice to consider why and how these metrics are different to give a better understanding.
Paper lacks any results about biases based on religion and race even though they claim to have done so in the initial section of the paper. Only one incomplete figure about religion is present which is hard to comprehend as it lacks any legend.
The results and figures need to be explained further. Explaining the ideal score for an automatic summarizer in the figures would have given a better idea. I did not really understand the dashed lines in the figure given as Baselines. An explanation with an example will go a long way in making things much more clear and bringing out the results effectively.
First it claims no work has been done on bias-ness in text summarization then it points out a previous work on structure biasness (Jung et al, 2019).
In the paraphrasing of articles, the authors do not show how good the paraphrased sentences are. This raises the concern that if the paraphrased version of sentences itself were curated in a manner that they did not include proper information. A human evaluation of a few paraphrased sentences would have cleared this issue. Also there are no examples from the synthetic dataset curated by the authors.
In the Fig.3, two of the extractive summarisers are not mentioned namely presumm and matchsum


**Summary Of The Paper:**

The paper draws attention to the problem of Bias Analysis in text summarization tasks which has not been addressed before. It defines the different types of biases present in summarisation task. It performs Bias Analysis using both abstractive and extractive automatic summarising models. The experiments suggest biasness depicted by these models

**Summary Of The Review:**

The paper lacks comprehensive evaluations using different embeddings, models and datasets. Further, the paper can be hard to comprehend due to lack of explanations of figures and curated examples.

============After rebuttal============================

The authors acknowledge the limitations pointed out, however the paper is very rushed and incomplete in terms of experimentation and their results. Adding the suggested results would definitely improve the quality of the paper, therefore I encourage the authors to do. However, it is still a reject from me with no changes in the previous score.

---

> ### Author Response · Authors · 2022-11-18
> **Author Response**
>
> Thank you for your detailed comments and suggestions.
>
> > The paper does not compare the results of automatic summarizers with human curated summarization.
>
> We do present this comparison for the two types of structure bias we analyze. However, you are correct that we don't provide these results for content bias. This is because, unlike the two other types of of bias, underrepresentation doesn't depend on bias amplification relative to human summaries, but amplification relative to the original article. We do not make the assumption that the human summaries are the gold standard in terms of bias (as this has not been established), and instead compare the ratios of information to the those in the article. However, we recognize that also including human summaries could be another interesting point of comparison.
>
> > Only used single embedding word2vec (word embeddings can themselves be baised)
>
> Yes, word embeddings themselves can be biased, and we see some evidence of biased associations with groups. We use thresholds to discard words with low similarities which we find mitigates some of this effect. Our hope is partially to exploit the bias in word embeddings as we know they correspond to human biases and may allow us to better capture when a group is being talked about through implicit reference (which keyword only approaches cannot capture). Nonetheless, this is a limitation that we acknowledge.
>
> > To add value, authors should have explored other text summarization datasets with different writing styles to really bring out the different style biases rather than just paraphrasing.
>
>  Other datasets may offer some insight into style bias, but different types of style bias what we explore. Our goal is to measure if perturbations to the style within a document affect a generated summary. Changing the style of the entire document and measuring what happens is a different question, and would be interesting for future work.
>
> > The paper lacks any association/description between the metrics used for bias in this paper v/s different text summarisation metrics. It would have been nice to consider why and how these metrics are different to give a better understanding.
>
> Typical evaluation metrics for summaries are designed to approximate summary quality by measuring overlap with some gold standard summary. This is a very different task from our goal of measuring bias, so we consider this comparison out of scope for this work. Here our goal is to measure how well represented groups are in summaries, regardless of the quality assigned by any evaluation score. Note that it is possible for a summary to be considered well written while still containing bias, and equally possible to have equally represented groups while having a summary that is essentially useless. As with classification, fairness is a different measurement than performance.
>
> > Paper lacks any results about biases based on religion and race even though they claim to have done so in the initial section of the paper.
>
> Descriptions of results for religion and race are in our appendix. There are some problems with measuring group membership for race, which is also discussed there. We have clarified our claims at the start of the paper.
>
> > * Only one incomplete figure about religion is present which is hard to comprehend as it lacks any legend. The results and figures need to be explained further.
> > * Explaining the ideal score for an automatic summarizer in the figures would have given a better idea. I did not really understand the dashed lines in the figure given as Baselines. An explanation with an example will go a long way in making things much more clear and bringing out the results effectively.
> > * First it claims no work has been done on bias-ness in text summarization then it points out a previous work on structure biasness (Jung et al, 2019).
>
> Thank you for the comment. We have added clarifications to all of these points. To further clarify the last one, while there is previous work for position bias (Jung et al, 2019), there is not another work we are aware of that considers this type of bias in the context we do, which includes bias towards specific groups as well as toward structure of articles.
>
> > In the Fig.3, two of the extractive summarisers are not mentioned namely presumm and matchsum
>
> PreSumm and MatchSum are missing because we cannot modify the length of summary generated by pretrained models. In Fig. 3 we consider summaries 50% the length of the original article. We have clarified this in the figure caption.

---

### Official Review · Reviewer_tjG5 · 2022-10-25

**Confidence:** 3
**Correctness:** 2
**Technical Novelty And Significance:** 2
**Empirical Novelty And Significance:** 2
**Recommendation:** 3

**Clarity, Quality, Novelty And Reproducibility:**

The work well-motivated and addresses a relevant problem in language generation. However, the methods lack clarity in some respects (like the paraphrasing approach) and some conclusions are not strongly supported by empirical evidence (see weaknesses 1 and 2 above). The authors mention the release of dataset and code, which helps in the reproducibility.

**Strength And Weaknesses:**

Strengths:
1. The paper addresses an important and underexplored topic in the automatic summarization.
2. The experiments show that under similar conditions of positioning in the source documents, summarization systems demonstrate preference for groups, for instance, by amplifying content related to men (versus women).

Weaknesses:
1. The content bias analysis is based on a synthetic dataset generated by GPT-2, so that the summary inference was performed on out-of-domain data. If summarization models are sensitive to paraphrasing and structural features, then synthetic documents may also exhibit artifacts that influence the results. It would be important to perform a similar analysis on selected documents from CNN/DM. Alternatively, a human evaluation experiment using the synthetic data could provide more evidence that the bias is really introduced by the summarization models.
2. The position bias analysis, the authors state that "clearly amplified by MatchSum, PreSumm, and Azure, as shown in Fig. 4b." However, it is not clear if the observed pattern is caused by model preference or simply by truncation of inputs. If we observe the figure in Appendix E, the distribution for PEGASUS and BART are more similar to the reference summaries distribution. In contrast, MatchSum, Presumm, and Azure decrease the frequency sharply between 0.4 and 0.6. Interestingly, Presumm and MatchSum use the same backbone model with a maximum input length of 512 tokens, whereas PEGASUS and BART support inputs of 1024 tokens, which might explain their capacity to select more content from the later positions of the articles. For reference, CNN/DM input documents have 766 words on average ("Extractive Summarization as Text Matching", Zhong et al., 2020). Do the quantile calculation in the figures take into account this truncation effect?
3. No details or citation is provided for the paraphrasing model. Just the link for its checkpoint at https://huggingface.co/tuner007/pegasus_paraphrase.
4. Authors mention that they "conduct manual analysis on a randomly selected subset of the article" but no further details are provided.
5. Experiments are based only on CNN/DM, which is well known for its lead bias for important content. Experiments on additional datasets could make the claims stronger.

Minor comments:
1. Typo in page 6: "Overall, t hese results show a pattern of unpredictability..."

**Summary Of The Paper:**

This paper investigates the presence bias introduced by text summarization models. The authors propose to measure bias in two dimensions: content and structure. They define content bias as tendency to mention a specific group (e.g., gender, religion, etc) in a text, whereas structural bias refers to bias as a result of structural features of the text (position, sentiment, and style). In this paper, they limit the scope of investigation to one type of content bias (underrepresentation) and two types of structure bias (position and style). To measure content bias, a representation score R(T, g) is proposed to gauge the proximity of a group g to a text T. For position bias, the content of model-generated summaries and reference summaries is compared according to its position in the input documents. Finally, for style bias, the authors measure the impact of paraphrasing sentences in the original document, under the assumption that the same content should be selected regardless of the style of writing. Those criteria were applied to a synthetic dataset generated by GPT-2 and the CNN/DM news summarization dataset. Based on the results, the authors conclude that summarization models exhibit preference for certain groups over others, amplify patterns of bias, and demonstrate sensitivity to the structure of articles.

**Summary Of The Review:**

The claims that summarization models exhibit content bias are weakly supported by the experiments. In terms of structural bias, this paper finds that summarization models trained on news articles have lead bias, which is a well-known fact in the literature. For the lack of novelty and solid evidence for the claims, the recommendation is for the paper rejection.

---

> ### Author Response · Authors · 2022-11-18
> **Author Response**
>
> Thank you for your comments and suggestions.
>
> > The content bias analysis is based on a synthetic dataset generated by GPT-2, so that the summary inference was performed on out-of-domain data. If summarization models are sensitive to paraphrasing and structural features, then synthetic documents may also exhibit artifacts that influence the results. It would be important to perform a similar analysis on selected documents from CNN/DM. Alternatively, a human evaluation experiment using the synthetic data could provide more evidence that the bias is really introduced by the summarization models.
>
> The concerns about out-of-domain data are valid. Our goal in using the synthetic data was to control the amount of group specific information present in articles, which is very hard with real data. Generating articles allows us to control for the impact of having different proportions of information about men vs women, for example, as well as the order it appears in. This is not possible with real data. Nevertheless, we acknowledge that the possibility of this data being out-of-domain is an inherent limitation.
>
> > The position bias analysis, the authors state that "clearly amplified by MatchSum, PreSumm, and Azure, as shown in Fig. 4b." However, it is not clear if the observed pattern is caused by model preference or simply by truncation of inputs. If we observe the figure in Appendix E, the distribution for PEGASUS and BART are more similar to the reference summaries distribution. In contrast, MatchSum, Presumm, and Azure decrease the frequency sharply between 0.4 and 0.6. Interestingly, Presumm and MatchSum use the same backbone model with a maximum input length of 512 tokens, whereas PEGASUS and BART support inputs of 1024 tokens, which might explain their capacity to select more content from the later positions of the articles. For reference, CNN/DM input documents have 766 words on average ("Extractive Summarization as Text Matching", Zhong et al., 2020). Do the quantile calculation in the figures take into account this truncation effect?
>
> It's true that truncation has an effect on these plots. There are some articles that are too long for MatchSum and Presumm to summarize without truncation. This does explain the dropoff you mention between 0.4 and 0.6. However, if these models were selecting text from throughout the articles rather than simply from the beginning, we would expect to see higher amounts of sentences selected from between 0.1 and 0.4 than in the models with longer limits, which we don't see. Instead, we see far more sentences being selected from before 0.1, meaning that even accounting for truncation, MatchSum and Presumm still select more sentences from the very beginning of articles than other models. We have clarified this point in the appendix.
>
> > Authors mention that they "conduct manual analysis on a randomly selected subset of the article" but no further details are provided.
>
> In our analysis we find that there are articles with cohesive structures that seem to align with human written articles. Among these articles, we observe that there's often a lack of semantic coherence (anaphoric references that don't align with what's written, contradictions, etc.). While this is a limitation of these articles, our hope is that the cohesive structure is sufficient to make up for these problems. We release our data to allow others to also analyze our generated articles.
>
>
> > Experiments are based only on CNN/DM, which is well known for its lead bias for important content. Experiments on additional datasets could make the claims stronger.
>
> It is true that CNN/DM contains lead bias. However, our results show some summarizers amplifying this lead bias (e.g. MatchSum) or largely ignoring it and selecting from later in the article (GPT-3). We agree that repeating this on further datasets with different position biases would add weight to what we show here, but our results still show that some summarizers amplify/follow existing bias while others largely ignore it.

---

### Decision · Program_Chairs · 2023-01-20

**Decision:**

Reject

**Justification For Why Not Higher Score:**

n/a

**Justification For Why Not Lower Score:**

n/a

**Metareview: Summary, Strengths And Weaknesses:**

The reviewers, quite in consensus, felt this paper is not ready yet for publication. There were concerns about correctness and the evaluation as compared to previous work, and the contribution itself, even if correct, being sufficiently significant. The authors did not respond to all reviewer comments in the second round.